# Structural basis for recognition and regulation of arenavirus polymerase L by Z protein

Huiling Kang[1,2,5], Jingyuan Cong[1,3,5], Chenlong Wang[1,3,5], Wenxin Ji[1,3], Yuhui Xin[1,3], Ying Qian[1], Xuemei Li[1✉], Yutao Chen [1✉] & Zihe Rao [1,4]

Junin virus (JUNV) causes Argentine hemorrhagic fever, a debilitating human disease of high mortality rates and a great risk to public health worldwide. Studying the L protein that replicates and transcribes the genome of JUNV, and its regulator Z protein should provide critical clues to identify therapeutic targets for disrupting the life cycle of JUNV. Here we report the 3.54 Å cryo-EM structure of the JUNV L protein complexed with regulator Z protein. JUNV L structure reveals a conserved architecture containing signature motifs found in other L proteins. Structural analysis shows that L protein is regulated by binding of Z protein at the RNA product exit site. Based on these findings, we propose a model for the role of Z protein as a switch to turn on/off the viral RNA synthesis via its interaction with L protein. Our work unveils the mechanism of JUNV transcription, replication and regulation, which provides a framework for the rational design of antivirals for combating viral infections.

[1] National Laboratory of Biomacromolecules, Institute of Biophysics, Chinese Academy of Sciences, Beijing, China. [2] State Key Laboratory of Biotherapy, West China Hospital, Collaborative Innovation Center for Biotherapy, Sichuan University, Chengdu, China. [3] University of Chinese Academy of Sciences, Beijing, China. [4] Laboratory of Structural Biology, School of Medicine, Tsinghua University, Beijing, China. [5] These authors contributed equally: Huiling Kang, Jingyuan Cong, Chenlong Wang ✉email: lixm@ibp.ac.cn; chenyutao@ibp.ac.cn

Arenaviruses comprise a group of viruses that primarily infect rodents, thus establishing life-long chronic infections. Infected rodents serve as natural reservoirs of viruses. Certain arenaviruses are known to infect humans, causing serious illnesses and even death. They include the Old World arenaviruses like Lassa mammarenavirus (LASV), the New World arenaviruses like Machupo mammarenavirus (MACV) and Junin virus (JUNV) that are known to cause zoonotic diseases[1,2]. JUNV causes Argentine hemorrhagic fever, which is characterized by mortality rates as high as 30% when left untreated[3]. Humans are commonly infected incidentally when in contact with secretions emanating from infected rodents[4,5]. While JUNV is currently endemic to South America, it is likely to swiftly spread to Europe, North America, and Asia, due to the rapid flow of people and goods[6,7].

The genome of JUNV consists of two ambisense RNA segments named long (L) strand (7.2 kb) and short (S) strand (3.5 kb) based on the length. These RNAs encode four viral proteins, namely GPC, NP, Z, and L proteins. The envelope glycoprotein precursor (GPC) and the nucleocapsid protein (NP) are encoded by the S strand, while matrix zinc-binding protein (Z) and the large protein (L) encompassing the RNA-dependent RNA polymerase (RdRp) are encoded by the L strand[6,8].

The L protein plays crucial roles in the processes of replication and transcription during the life cycle of the virus[8,9]. Therefore, L protein is an ideal target for the development of antiviral drugs, as these drugs would be able to specifically target virus protein instead of human cells. In addition, the chances of developing drug resistance are relatively low because the L structures and their amino acid sequences are highly conserved, indicating that these features are critical for proper viral function[10]. Lastly, these structures are conserved across virus species, raising the possibility of the development of broad-spectrum drugs. Until recently, little detail was known about the structure of the L protein harboring the RdRp of arenaviruses, where structures of LASV and MACV L proteins have been elucidated with cryo-electron microscopy[11,12].

Z protein participates in several stages of the viral life cycle and plays essential roles in: (i) regulating the synthesis of vRNA by interacting with L protein, (ii) modulating viral assembly and budding by interacting with GP as well as RNPs and driving particle release at the plasma membrane, (iii) interacting with host cell proteins, and (iv) antagonizing interferon[13–15]. Although studies have shown that Z protein interacts with L to affect the process of replication and transcription, details of how Z protein binds to L and regulates its functions are unknown to date[16–19].

In this report, we set out to elucidate the cryo-EM structure of the complex of L with Z protein from JUNV. Our structural analyses provide critical insights into the domain organization of L, the nature of its active site as well as key catalytic residues, and map the precise location of the Z protein binding site on the L protein.

## Results

### Determining the structure of JUNV L–Z complex by cryo-EM.
We expressed JUNV L protein and Z protein separately in *sf*9 insect cells using a Bac-to-Bac expression system. The recombinant proteins were purified by strep-tag and His-tag based affinity purification. In order to form a stable L–Z complex, purified L and Z proteins were mixed at a molar ratio of 1:2 and incubated for 40 min on ice. Analysis of the purified L–Z complex using SDS-PAGE showed that the complex contained the full-length L protein and Z protein (Fig. 1a, b).

For cryo-EM data collection, we used a 300 kV FEI Titan Krios transmission electron microscope with a GIF-Quantum energy filter (Gatan) and a Gatan K2-summit detector (Fig. 1c, d). An initial 3D model was reconstructed from 2,184 micrographs, at 4.27 Å resolution. To improve the resolution, we merged 6,409 micrographs from three separate datasets, and obtained a final 3.54 Å map, which we refined further by incorporating 362,657 particles (Fig. 1e and Supplementary Fig. 1). Statistics for cryo-EM data collection are summarized in Supplementary Table 1.

The model was built ab initio using the structure of MACV L protein (PDB entry: 6KLD, amino acid sequence identity of 73% with JUNV L protein) as a guide and refined by PHENIX[20] real-space refinement, together with rounds of manual adjustment in COOT[21]. The final structures were validated using CCP4[22] and PHENIX[20], with an r.m.s.d. of Cα of 0.95 Å in comparison to MACV L protein. The density of the map showed that the complex structure contains a characteristic ring-like core RdRp region and the interface between L protein and Z protein. However, we found that the C-terminal region (~400 amino acids, spanning residues 1818–2210) is missing in the electron density map, and a few gaps within the connecting loops can not be modeled due to ambiguity in the regions[10,23]. For Z protein, only the middle region (AA 31–82) was identified and modeled in the observable density map, while N-terminal (AA 1–30) and C-terminal (AA 83–94) regions were not modeled due to the flexible nature and missing density (Supplementary Table 2). Statistics of structure determination are summarized in Supplementary Table 1.

### Overall structure of the JUNV L–Z complex.
JUNV L protein, like other arenavirus L proteins, is composed of a single polypeptide chain; this chain can be divided into three parts, similar to the structural arrangement of influenza virus RNA polymerase complex[24–27]. Therefore, we designated these three parts starting from the PA-like region at the N-terminal, RdRp region in the middle and PB2-like region at the C terminal (Fig. 2a and Supplementary Fig. 2). The PB2-like region of JUNV L contains the electron density for only the thumb ring and the lid domain. Our analysis showed that the RdRp region is located at the core of the structure, while both the C-terminal of PA-like region as well as the N-terminal of PB2-like region tightly bind the thumb domain of RdRp region. On the other side of the RdRp region, the endonuclease region binds both the finger domain and the α-Ribbon domain. The full-length Z protein can be divided into three parts: N-terminal region, middle RING domain region, and a C-terminal region, as previously described[28]. In our structure, the RING domain is bound to the L protein. The Z protein, as an appendage of L protein, bound with the core-lobe domain of the PA-like region as well as the palm domain of the RdRp region on the opposite side of the endonuclease. The structure of Z protein contains a pair of anti-parallel β-strands and one α-helix (Fig. 2b–d). One zinc finger (C41, C42, C58, C61) involving four cysteines is observed. Another plausible zinc finger was found in the vicinity of the former, including three cysteines and one histidine (C52, C72, C75, H55). However, the distribution and orientation of the residues were distorted in the structure, and we did not identify the density for zinc ion. Incidentally, the Z protein from LASV has two zinc fingers[29] (Supplementary Fig. 3).

### Polymerase active site of JUNV L protein.
The JUNV L protein catalyzes replication and transcription of the viral genome. Structural analysis showed that the active site at the RdRp region and motif organization is highly conserved in arenaviruses such as the MACV L protein[11]. JUNV L active site contains eight conserved motifs (motifs A–H), including six conserved polymerase motifs (A-F) and two specific motifs (G and H) found in

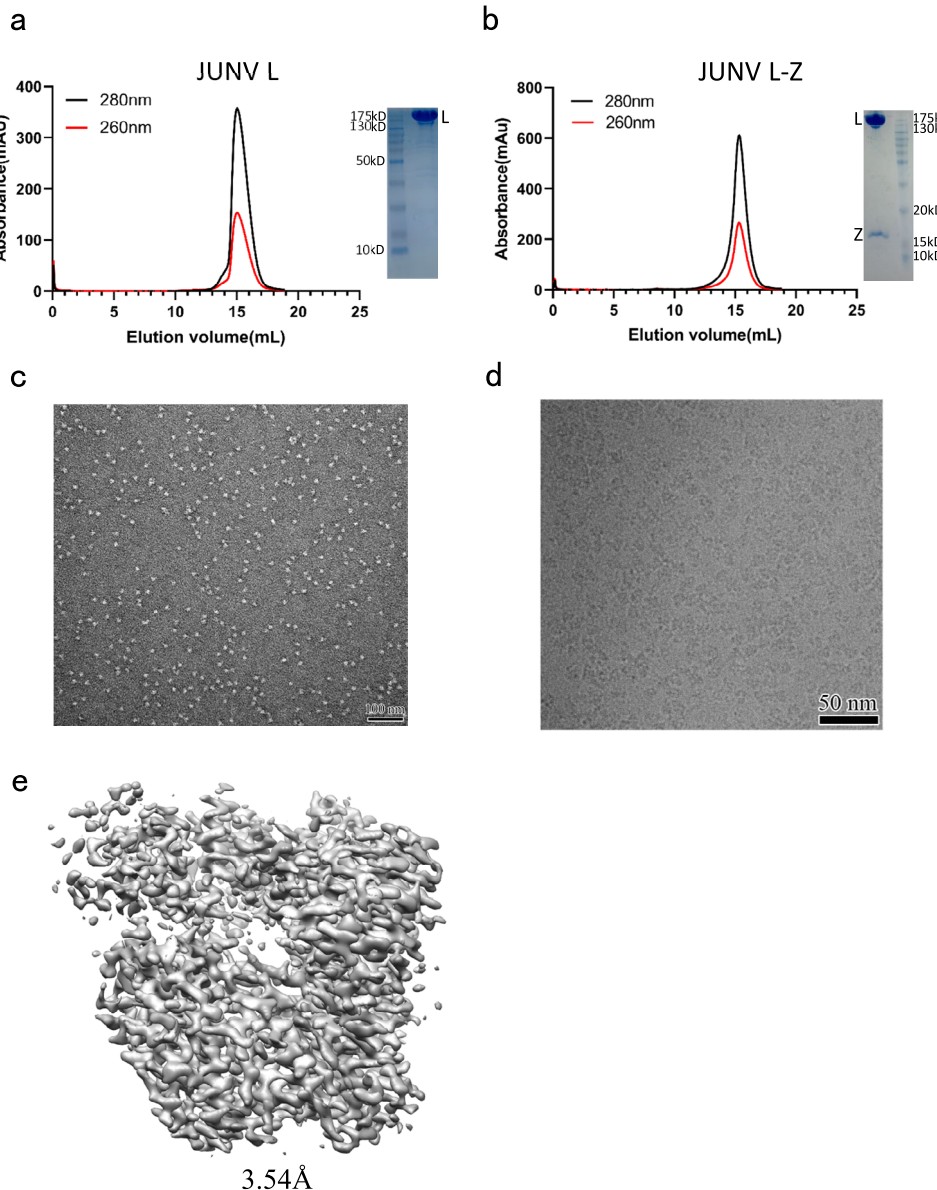

**Fig. 1 Biochemical characterization and cryo-EM structure. a** Elution profile of the purified JUNV L protein on a Superose 6 Increase 10/300 GL size-exclusion column. SDS-PAGE was performed to assess the quality of L protein (repeated ≥ 3 times). **b** Elution profile of the purified JUNV L–Z complex on a Superose 6 Increase 10/300 GL size-exclusion column. SDS-PAGE was performed to assess the quality of L–Z complex (repeated ≥ 3 times). **c** A representative negative staining micrograph of JUNV L–Z complex (repeated ≥ 3 times). **d** A representative cryo-EM micrograph of JUNV L–Z complex (out of ~6400 micrographs). **e** The final cryo-EM density map of JUNV L–Z complex.

segmented negative-stranded RNA virus (sNSV) polymerases (Fig. 3a, b)[30,31]. Each of these motifs contains a characteristic conserved signature. For instance, residue D1187 of motif A binds one $Mg^{2+}$ ion, while motif B contains a conserved H1295. Motif D and motif E are in close contact, which is bridged by interaction with the $Mg^{2+}$ ion. Motif C (1327-SDD) is the catalytic motif, which is ubiquitous in other RNA polymerases and is located directly opposite to the position of motif F. Different from the other sNSV polymerases that need the presence of 5′-vRNA to stabilize its motif, the motif F of arenaviruses has two conserved residues, K1121 and R1128, that help localize the finger tip and the associated extension loops. This region of the JUNV polymerase is present in a highly ordered manner, similar to MACV and LASV L proteins[11]. The fingertip is accommodated in the space since the finger extension helix (AA 712–729) is oriented in an extended conformation. Moreover, the residues from the core-

lobe helices also form extensive interactions with residues in the fingertip, thus stabilizing its conformation. Together, these observations indicate that the active site of arenavirus polymerase may not require the 5′-vRNA to stabilize the fingertip conformation, enabling JUNV L to replicate and transcribe RNA in the presence of only 3′-vRNA, a feature that might be unique for arenavirus polymerase. The R635 of motif G is positioned to interact with the priming NTP. The K1233 of motif H stabilizes the motif B backbone conformation by hydrogen bonding to multiple carbonyl-oxygens (Fig. 3c, d). Together, these observations show that the JUNV L also contains specific sNSV polymerase motif G and motif H similar to that observed in the LACV L protein[31].

Next, we performed an in vitro RNA synthesis assay to evaluate the JUNV L catalytic function and regulation. L protein is catalytically active in the presence of 3′-vRNA, and is activated by

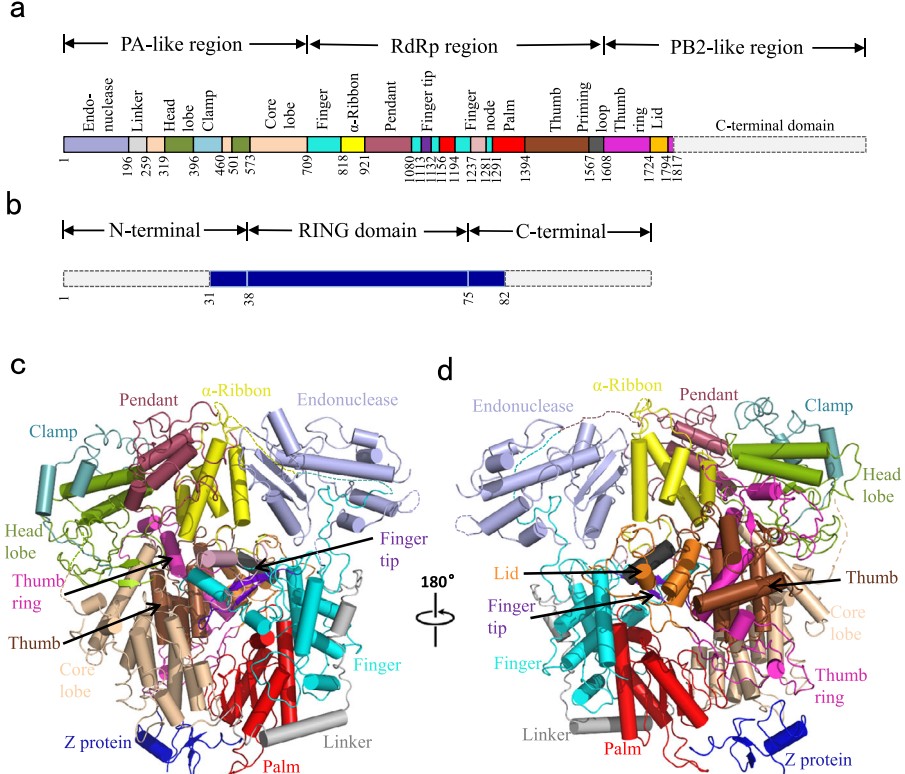

**Fig. 2 Overall structure of JUNV L–Z complex. a** Schematic diagram of the domain architecture of JUNV L protein divided into three parts: the PA-like region, the RdRp region and the PB2-like region. Each domain of JUNV L is represented using a unique color. The unresolved regions of L C-terminal are indicated by dotted lines. **b** Schematic diagram of the domain architecture of JUNV Z protein divided into three parts: the N-terminal, the middle RING domain and the C-terminal. The unresolved regions of N-terminal and C-terminal are indicated by dotted lines. **c**, **d** Cartoon representation of JUNV L–Z complex. The structures are colored by domains, the coloring scheme is identical to that in (**a**, **b**). The unresolved linker regions are connected by dashed lines. (**d**) Regions identical to those depicted in (**c**), but rotated 180° along the vertical axis.

the presence of $Mg^{2+}$ or $Mn^{2+}$, which is essential for the polymerase activity. Meanwhile, L could not be activated by $Zn^{2+}$ (Supplementary Fig. 4a). Our in vitro assay also showed clearly that in the presence of Z, the RNA synthesis activity of L will be inhibited (Supplementary Fig. 4b). To confirm the interaction of L protein with RNA, we performed an Electrophoretic Mobility Shift Assay (EMSA). Both L and L–Z complex can bind individually to the 3′-vRNA while the length of RNA has a profound effect on the binding affinity (Supplementary Fig. 4c), indicating that the L-binding region resides in the terminal 40 nt of the 3′-vRNA.

**Interactions between JUNV L and Z protein.** The Z protein is known to bind and inactivate L protein in arenavirus, a phenomenon that was reported previously[15] and also verified in our in vitro experiment. To reveal the detailed interaction location and binding mode, we obtained JUNV L–Z complex and analyzed its cryo-EM structure. The core-lobe domain of JUNV L protein interacts with the RING domain of Z protein. In addition, the palm domain of L protein is also in contact with the Z protein (Fig. 4a and Supplementary Fig. 5a, b). Our analysis showed that upon complex formation, L and Z proteins buried a total surface area of 2,150 Å². The interactions between the L and Z proteins occur in two distinct regions: (1) The first region involves extensive interactions between the residues S31, R36, W43 of the Z protein and the residues F688, V1177, N1179, T1376, F1377 of the L protein. The interactions between these residues include both hydrogen bonds and hydrophobic interactions. Residue W43 of Z participates in hydrophobic interaction and is flanked by F688 and F1377 from L protein on either side, where W43

formed strong aromatic stacking with F1377 of L protein. This interaction is probably crucial to stabilize the L–Z interaction and correct orientation of Z protein. Except for this hydrophobic interaction, other interactions in this region are dedicated by hydrogen bonds (Fig. 4b and Supplementary Fig. 5c). (2) The second region of interaction is between the C-terminus of the RING domain and a loop (AA 677–688, designated as "680 loop" in the following sections) from L protein, including residues C41, R60, V64, M65, N73 from Z protein and residues E682, L686, Y687 from L protein. Hydrophobic interactions were found between C41, V64, M65 of Z protein and L686 of the L protein. The remaining interactions between L and Z are comprised of hydrogen bonds (Fig. 4c and Supplementary Fig. 5c). Sequence alignment of the residues from JUNV L protein involved in binding Z protein reveals that these resides are conserved in other arenaviruses[17](Supplementary Fig. 5d). Furthermore, Z protein fits in the cavity of L protein like a lid, and the contact surfaces of the L and Z proteins comprise opposite charges, suggesting that electrostatic interactions also play a critical role for the formation of the complex (Fig. 4d).

To confirm the location of the interaction interface observed in the structure, we performed MicroScale Thermophoresis (MST) assays (Supplementary Fig. 6). First, we synthesized two peptides, Z (AA 18–47) and Z (AA 48–78), corresponding to the two regions of Z protein that bind the L protein. We found that the Z peptides/protein interact with L, albeit with varying affinities. We determined the equilibrium dissociation constant ($K_D$) of different lengths of Z with full-length L to be $393 \pm 102$ nM for Z (full length), $863 \pm 220$ nM for Z (AA 18–47) and $1.15 \pm 0.45$ μM for Z (AA 48–78). In order to validate the role of key residues at the

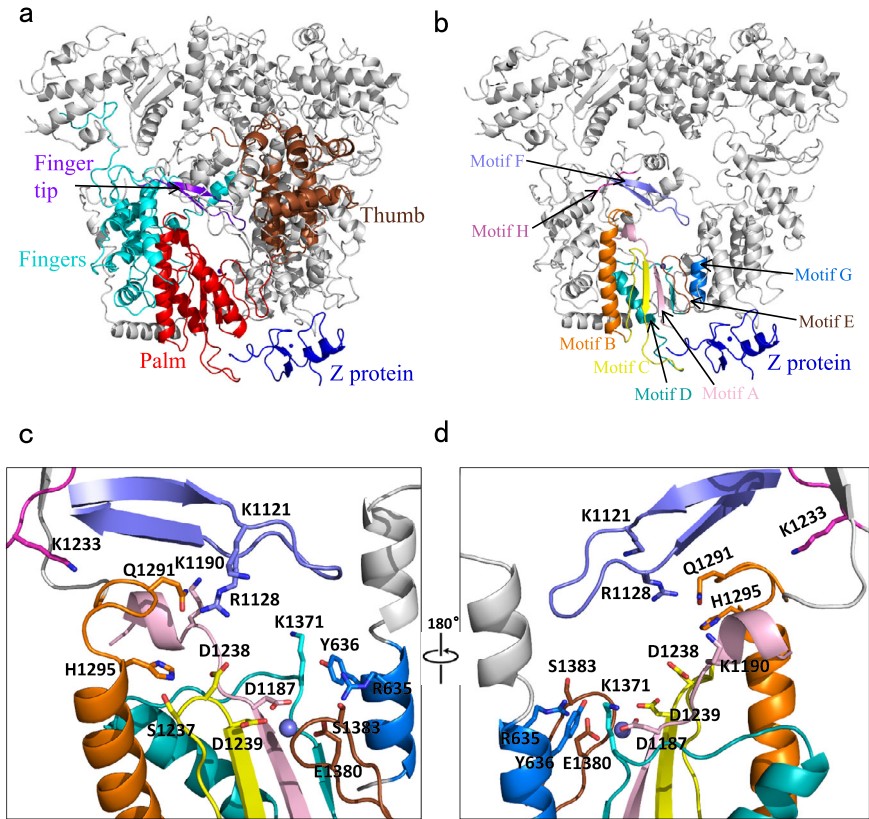

**Fig. 3 Structure of the JUNV L RdRp region. a** The RdRp region is highlighted in colors: the finger is shown in cyan, the thumb in brown, the palm in red and the finger tip in purple. The Z protein is colored in blue and the remaining structure of L is colored in gray. **b** Similar view as shown in (**a**), but motifs A–H are highlighted in unique colors. The Mg atom is shown as a purple sphere, and the Zn atom is shown as blue sphere. **c**, **d** Magnified views of the active site, colored as in (**b**). Residues selected are shown as sticks, with oxygen atoms shown in red, nitrogen atoms shown in blue. The Mg atom is shown as a purple sphere. **d** Region identical to that in (**c**), but rotated by 180° along the vertical axis.

binding interface, we designed three Z mutants (R36A, W43A, R36A&W43A) and five L mutants (F688A, F1377A, VNN (1177–1179)AAA, Y687A&F1377A, F688A&F1377A) to test the affinity of mutants with a native binding partner. We determined the dissociation constants between L-*wt* and Z mutants to be 297 ± 59 nM (Z-R36A), 287 ± 53 nM (Z-W43A), and 318 ± 51 nM (Z-R36A&W43A). The results of the test for Z mutants indicated that either single mutants or two points mutants of Z failed to disrupt the binding between L and Z. In addition, we found that the L mutants bound with Z-*wt* at a dissociation rate of 1.77 ± 0.35 μM (L-F688A), 711 ± 162 nM (L-F1377A), and 914 ± 194 nM (L-VNN (1777–1779)AAA) with Z. These mutants showed a certain decrease of affinity, although the extent is not significant. Next, we tested the effect of L double mutations on Z-binding affinity. As shown in Supplementary Fig. 6j, k, L-F688A&F1377A and L-Y687A&F1377A have no observed binding affinity with L. Therefore we speculated that residues Y687, F688, and F1377 were important in the L–Z interaction, possibly in a synergistic manner, as indicated by the double mutations.

## Discussion

The structure of JUNV L protein in complex with its regulator of Z protein revealed the interaction details at 3.54 Å resolution. The L protein showed a conserved RdRp region across sNSV. In vitro RNA synthesis assay showed that the JUNV *apo* L protein has RNA polymerase catalytic activity, which could be inhibited by Z protein. Our structural analysis reveals that except for motif F, all other motifs perform similar functions as those observed for other sNSV polymerases. This observation explains why

arenaviruses exhibit polymerase activity only in the presence of 3′-vRNA, which is unique compared to other sNSV polymerases. Furthermore, the EMSA indicated that 3′−40 nt vRNA was capable of binding the L protein, and the presence of Z does not affect the binding of L protein with vRNA.

Our structure revealed that the L protein harbors a zinc finger structure, composed of residues C284-C287-C469-H471 located in the core-lobe domain (Supplementary Fig. 7a). We speculated that it might stabilize the 3′-vRNA template which has left the active site after RNA synthesis is initiated[11,31–33]. Our analysis suggests that 3′-vRNA can fit nicely into the highly positively charged channel of JUNV L protein by modeling the structure of MACV−3′-vRNA over JUNV L structure (Supplementary Fig. 7b–d).

Since experimental evidence describing the precise location of template RNA entrance/exit, NTP entrance, and nascent RNA exit remains absent, our assumptions regarding these locations are based on the charge distribution within the internal channel, and comparison with the LACV L protein (Fig. 5a, b). And similarly, we also noticed a pendant domain near the putative template entrance, which is different from the other sNSV polymerases and may regulate the transcription/replication activity.

By sequence and structural analysis of JUNV L–Z complex we identified amino acids that mediate the interaction between L and Z proteins. The amino acids of Z protein responsible for binding L are conserved in arenaviruses (Supplementary Fig. 5e). A key residue, W43, of Z protein sandwiched between two phenylalanines (F688 and F1377) of L protein, is conserved in Tacaribe virus (TCRV) (W44) and lymphocytic choriomeningitis virus

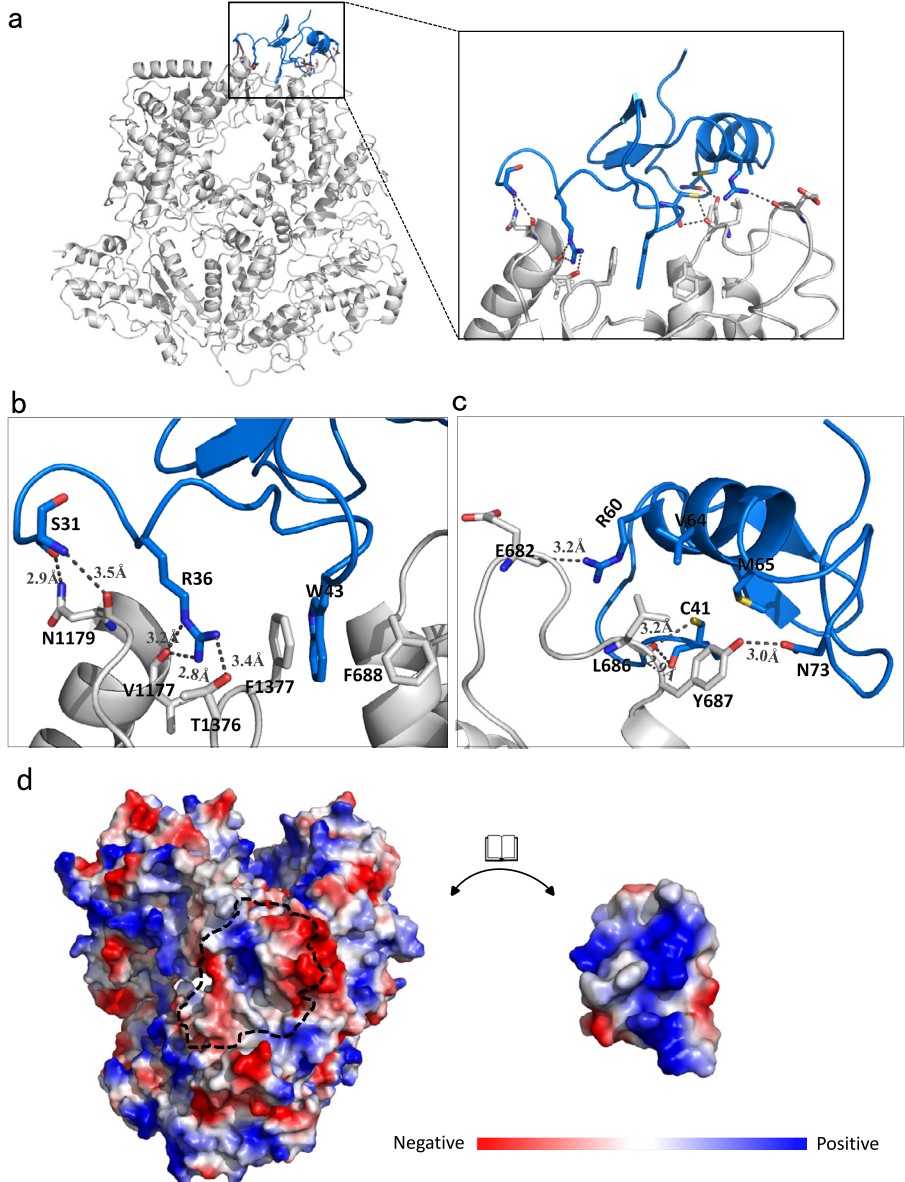

**Fig. 4 Details of the molecular interactions between JUNV L and Z protein. a** A cartoon representation of JUNV L–Z complex structure is shown, with L shown in gray and Z in blue. The black boxes indicate the magnified view of the interactions between L and Z. **b, c** Magnified views of interaction interfaces between L and Z. The representative residues involved in the interaction are shown, main chain or side chains involved in hydrogen bonds or hydrophobic interactions are shown as sticks, with oxygen, nitrogen and sulfur atoms shown in red, blue and yellow, respectively. Hydrogen bonds are depicted as black dashed lines. **d** L and Z are shown as molecular surfaces colored by electrostatic potential from red to blue, negative to positive, respectively. The surface of each protein is shown in an open-book representation. The approximate binding interface is outlined by a black dashed line.

(LCMV) (W36). Previous studies on TCRV and LCMV showed that this tryptophan is important in the interaction between L and Z[16,34]. This finding is consistent with our structure described here. Moreover, residues Y37, L49, and Y57 of JUNV Z protein, also conserved in TCRV and LASV, might engage in hydrophobic interactions to stabilize the structure of Z protein. Structural analysis also revealed that the 680 loop (AA 678–689) of MACV L sterically hinder the interaction of L with the Z protein. On the basis of these findings, we hypothesize that the 680 loop (residues 677–688) of JUNV might be a crucial molecular switch that regulates the assembly and disassembly of the L–Z protein complex. By conformational adjustment, the L protein either binds to or dissociates from the Z protein, in line with specific stages of infection. For example, shortly after viral entry into the host cell, a conformational change of the 680 loop could

destabilize the interaction between L and Z, activating the L protein to begin RNA synthesis. During the late stage of viral particle assembly, in contrast, the 680 loop is assumed to undergo a conformational change that facilitates its binding with Z protein, thus inactivating the L protein (Fig. 5c). Interestingly, multiple sequence alignment of the 680 loop revealed that only part of the amino acids (I677, L678, and A684) is conserved (Fig. 5d).

Previous studies on the L protein from LCMV[35] and MACV[11] suggest that dimerization can facilitate the activity of polymerase and the presence of 3′-vRNA facilitates the dimerization of the L protein. Intriguingly, the JUNV L proteins are present in monomeric form as shown by gel-filtration and cryo-EM classification, and our attempts to incubate JUNV L protein with 3′-vRNA did not induce dimerization. We therefore superimposed

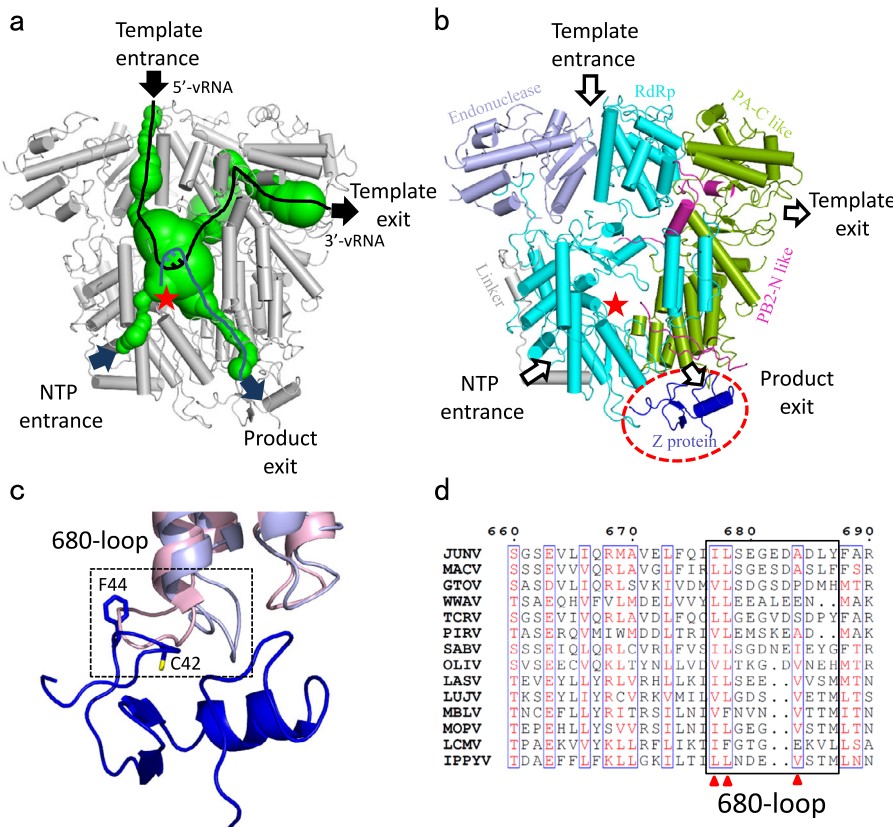

**Fig. 5 Schematic diagram showing features of JUNV L protein for RNA synthesis. a** Tunnels within the JUNV L structure for RNA synthesis. The template entrance and exit (black), the NTP entrance and product exit (blue) are shown. The catalytic site is indicated by a red star. **b** The product exit is blocked by Z protein. The protein structures are shown as cartoon and colored by domains, the catalytic site is indicated by a red star. The template entrance, NTP entrance, template exit and product exit are indicated by arrows. **c** Comparison of the 680-loop of MACV L protein and JUNV L–Z complex. Superimposition of the *apo* MACV L protein on the JUNV L–Z complex. The black dot box indicates magnified views of the 680 loops of the JUNV L protein (shown in light purple) or MACV L protein (shown in light pink), and displays the steric hindrance between MACV L and JUNV Z protein (shown in blue). **d** Sequence alignment of the region around the 680-loop. The conserved residues were indicated by red triangles. GTOV, Guanarito virus; WWAV, Whitewater Arroyo virus; PIRV, Pirital virus; SABV, Sabia virus; OLIV, Oliveros virus; LUJV, Lujo virus; MBLV, Mobala virus; MOPV, Mopeia virus; LCMV, lymphocytic choriomeningitis virus; IPPYV, Ippy virus.

the L protein of JUNV on the dimeric MACV L protein, but found no steric hindrances. The crucial residues located at the dimer interface between the two L proteins are conserved as well. Therefore we deduce that lack of certain unknown host factors might have rendered the JUNV L dimerization unsustainable, causing L to stay as monomers in the solution[36] (Supplementary Fig. 8c–g).

Our analysis also found no observable electron density for the C-terminal of the L protein, indicating that the connection between the C-terminal domain and the rest of L protein has a very flexible nature. This lack of C-terminal in the sNSV polymerase structures was common across sNSVs[11], indicating that this C-terminal region might not participate in the entire processes of transcription and replication, instead it might function at specific stages of RNA synthesis. In the sNSVs, this region corresponds to the PB2 cap-snatching domain of the influenza and performs cap binding for host mRNA snatching. It is well established that all sNSVs snatch host mRNA and cleave it to create a short capped primer for the viral mRNA transcription, which is called "cap-snatching" mechanism[10,31]. Thus, this domain can be considered as the switch which is responsible for the activation of transcription to synthesize viral mRNA or switch to vRNA synthesis for assembling new viral particles. We superimposed our structure on the dimer of the MACV L protein that contains the density of the C-terminal, and compared the C-terminal sequences of the JUNV and MACV L protein

(Supplementary Fig. 8a, b). We found the joint between C-terminal and the rest of L protein is indeed flexible and prone to fracture. Furthermore, the C-terminal of JUNV L protein also has the characteristic triple aromatic residues (F1906, Y1970, and Y1972) which might be the crucial candidate residues involved in cap binding[11] (Supplementary Fig. 8f).

On the basis of our and previous findings, we propose a model for the regulation of JUNV L protein during the infection cycle (Fig. 6). In step 1, following viral entry into the host cell, the RNP is released into the cytoplasm. Here, L is bound by vRNA at template channels and Z at the product exit, rendering L into an inhibitory state[14,15,19,37]. In step 2, at low concentration in the cellular environment, Z dissociates from L[15]. The dissociation is probably synergistically triggered by a conformational change at 680-loop near the product exit, which might be caused by host factors *e.g.* eIF4E[38] binding with the RING domain of Z protein. The bound vRNA activates the synthesis of RNA by the L protein in the absence of Z. During step 3, viral RNA synthesis (transcription and replication) is initiated and the RNA is elongated, with continuous inflow of NTP from substance entry. This process is accompanied by the 5' of nascent RNA going through the product exit, and the 3' of template RNA going through the template exit[39]. In step 4, the nascent RNA is released into the cytosol for translation of viral proteins or replication of viral genomes, once RNA synthesis is finished. In step 5, at the late stage of JUNV infection, Z is synthesized in large amounts[14,15,40].

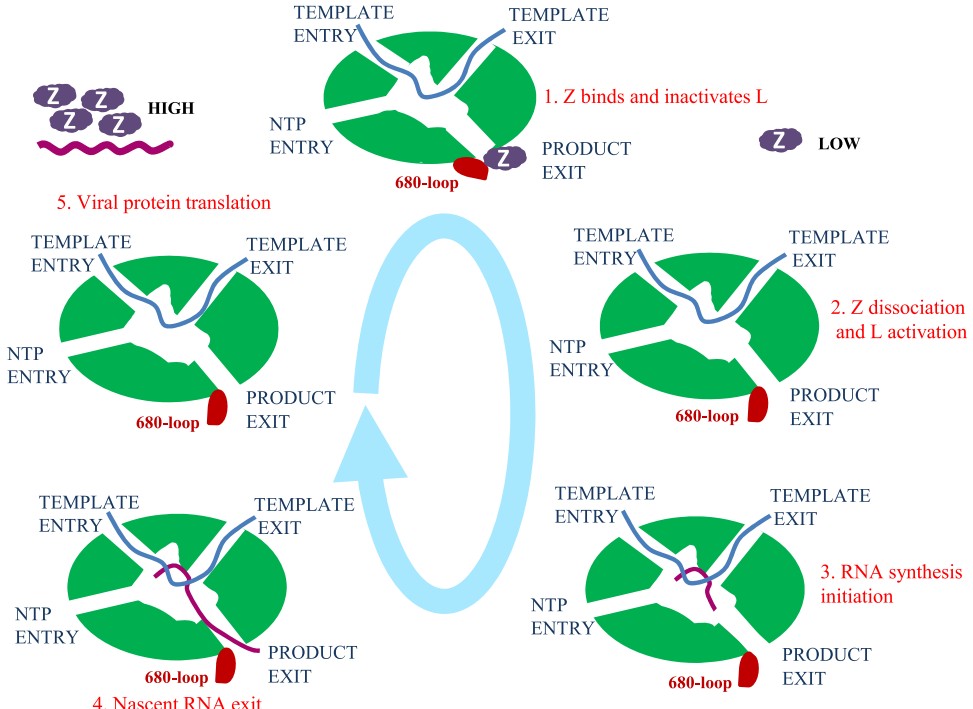

**Fig. 6 The proposed model of JUNV L regulation by Z during the viral life cycle and RNA synthesis.** The interaction and regulation of JUNV Z with L was divided into five separate steps, including: (1) Viral entry bringing inactivated L in RNP complex, (2) Z dissociation and L activation at low concentration of Z, 3) RNA synthesis initiation and elongation, (4) Nascent RNA's exit from L, (5) Z synthesized at late stage binds and locks L in inactivated state for virion packaging.

The Z protein at high concentration in the cytosol environment would bind to the 680 loop at the product exit and block the latter, locking L in an inactive state. This would signal the start of packaging newly synthesized and assembled RNPs into virions for the subsequent infection cycle[13,41].

In summary, we have determined the atomic resolution cryo-EM structure of the JUNV L–Z complex, and have elucidated the possible role of Z during the viral RNA synthesis. Our findings should facilitate a better understanding of the structural basis for arenavirus transcription and replication, regulation mechanism and evolution of the RNA polymerase amongst sNSVs. Our insights presented here should also assist in developing novel rational structure-based drugs to treat diseases associated with arenavirus infection.

## Method

**Protein expression and purification.** The codon-optimized sequences for JUNV L protein (GenBank: ABN11796.1) and Z protein (GenBank: ALE15111.1) (Supplementary Table 3) were subcloned into a pFastBac1 expression plasmid, and expressed in *sf*9 cells using a Bac-to-Bac expression system. The L proteins containing an N-terminal 2x Strep tag and C-terminal 6xHis tag containing Prescission protease cleavage site, as well as the Z proteins containing an N-terminal 6xHis tag were expressed in *sf*9 cells. Following culture for 72 h, cells were collected by centrifugation at 2,600 g for 20 min, before the cell precipitate was resuspended in lysis buffer (50 mM Tris-HCl pH 8.0, 500 mM NaCl, 10%(v/v) glycerol, 1 mM Tris (2-carboxyethyl)phosphine (TCEP) and EDTA-free protease inhibitor). After sonication, the cell lysate was centrifuged at 30,000 *g* for 40 min at 4 °C, the supernatant collected and applied to $Ni^{2+}$ -NTA agarose beads for protein purification. L and Z proteins were then eluted with a buffer containing 300 mM imidazole 50 mM Tris-HCl, 500 mM NaCl, 10% (v/v) glycerol and 1 mM TCEP. The eluted L and Z proteins were mixed and incubated for 1 h, followed by a separation step on Strep-Tactin XT to remove excess Z protein that failed to form a complex with L. After thorough washing with a buffer containing 50 mM Tris-HCl pH 8.0, 500 mM NaCl, 10%(v/v) glycerol, 1 mM TCEP, 1 mM EDTA, the complex of L protein and Z protein was eluted from the column using the same buffer plus 50 mM biotin. The eluted sample was applied to Superose 6 Increase 10/300 GL (GE Healthcare) to further purify the complex and then dialyzed into the same elution buffer without biotin or glycerol. The purified proteins and complex were flash-frozen in liquid nitrogen and stored at −80 °C before use.

**Electrophoretic mobility shift assay (EMSA).** In order to detect the interactions between L proteins and the vRNA promoter, four segments of JUNV were designed: (i) L-segment 3′−19 nt (5′ CGCCUAGGAUCCUCGGUGC 3′), (ii) L-segment 3′−30nt (5′ GAUUCCUCCAUGCUCAAGUGCCGCCUAGGA 3′), and (iii) L-segment 3′−40nt (5′ GAUUCCUCCAUGCUCAAGUGCCGCCUAGGAU CCUCGGUGC 3′). We incubated the 3′-vRNA with L protein at a molar ratio of 1:1.2 (L:vRNA) at 4 °C for 45 min, in the buffer of 50 mM Tris-HCl pH 8.0, 500 mM NaCl, 10% (v/v) glycerol, 1 mM TCEP. NativePAGE loading buffer (BN2003, Invitrogen) was added to the protein–RNA mixture and applied to NuPAGETM 4–12% Bis-Tris Gel (NP0322BOX, Invitrogen) with running buffer (2148468, Invitrogen) to examine the binding of L protein to vRNA.

**Micro-scale thermophoresis (MST) assay.** Prior to MST assay, proteins and peptides were dialyzed into a buffer containing 50 mM HEPES pH 7.8, 500 mM NaCl, 1 mM TCEP. L proteins (wild type and mutants) were labeled with fluorochrome NHS(L011) labeling kit. Wild type and mutants of Z proteins were each serially diluted twice to 16 concentration gradients, and then mixed with an equal volume of labeled L proteins (20–30 nM). The mixtures were incubated for five minutes at room temperature and aspirated into capillaries for measurement using Monolith NT.115 (Nano Temper Technologies). The dissociation constants were measured under 40% infrared laser power and at a constant temperature of 25 °C. The $K_D$ for labeled proteins and ligands was calculated using Nanotemper analysis software.

**Cryo-EM sample preparation and data collection.** The protein sample was eluted in a buffer containing 50 mM Tris-HCl pH 8.0, 500 mM NaCl, 1 mM TCEP. 4 μl of purified JUNV L and Z complex at 1 mg/ml concentration was applied to a glow-discharged amorphous alloy film (R1.2/1.3, Au, 300 mesh). The grids were blotted for 3 s at ~100% humidity and plunged into liquid ethane using an FEI Vitrobot Mark IV. The samples were recorded on a FEI Titan Krios transmission electron microscope operated at 300 kV with a GIF-Quantum energy filter (Gatan) and a Gatan K2-summit detector for data collection. All the cryo-EM movies were automatically collected using SerialEM software (http://bio3d.colorado.edu/ SerialEM/). The nominal magnification of ×215,000 corresponds to a calibrated pixel size of 0.65 Å at the specimen and a dose rate of 6e⁻/pixel/s. The total exposure time of each image was 4.2 s, to obtain an accumulative dose of ~60 e⁻/Å², fractioned into 50 frames. A total of 6409 images were collected. All images were recorded in a defocus range between −1.2 and −2.5 μm.

**Image processing.** The beam-induced motion and anisotropic magnification were corrected by the program MotionCorr2[42] with a 5 × 5 patch, and the initial contrast

transfer function (CTF) was estimated using the program CTFFIND4.1[43]. Images with resolution better than 4 Å were selected for subsequent data processing. We collected a total of 6409 images, of which 2,184 were selected for the calculation of the initial model. Using MACV L as the model to pick particles, approximately 253,000 particles were selected from 2184 micrographs for reconstruction. To process and reconstruct the structure, relion 3.0.8 was used[44]. The selected particles were divided into five classes through 3D classification. The best class containing 136,000 particles was selected for 3D refinement and obtained a density map with a resolution of 4.63 Å. To improve the resolution, CTF refinement and polish processes were used. As a result, a density map with a resolution of 4.27 Å was achieved. Following the merging of these 6409 images, approximately 1,255,000 particles were selected for 3D classification using the density map of 4.27 Å model (obtained previously) low-pass filtered to 15 Å as the initial model. After two rounds of 3D classification, a total of ~362,000 particles, accounting for 28.8% of the total number of particles were selected for further 3D reconstruction. After postprocessing, CTF refinement and polish processes, we got the final density map reaching a resolution of 3.54 Å as determined by the Fourier shell correlation (FSC) 0.143 cut-off value. The ResMap[45] was used to estimate the local resolution of the L–Z complex.

**Model building and refinement**. The 3.54 Å resolution map was used for model building and refinement. UCSF chimera[46] was used to position the coordinates of MACV L (PDB entry: 6KLD) into the map. Further adjustments, including optimization of the positioning of the side chains was performed in COOT[21]. We used PHENIX[20] to calculate the structure factors. The complete structure was subjected to global real-space refinement in multiple cycles using rotamer and Ramachandran plot restraints.

**In vitro enzymatic assay of JUNV L protein**. All naked RNA templates were chemically synthesized and purified (Ruibiotech). NTPs were purchased from Takara Biomedical Technology. $[\alpha\text{-}^{32}P]$-ATP and $[\gamma\text{-}^{32}P]$-ATP were purchased from PerkinElmer.

A standard polymerase assay was performed similar to that described elsewhere[11,47]. In brief, the 40-nt 3′-vRNA promoter was denatured at 70 °C for 10 min, and then immediately placed on ice. In order to obtain a strong enzymatic active signal, 1 μM of 3′-vRNA with 1 μM of JUNV L protein were used in RNA synthesis assay containing 20 mM Tris-base, pH 8, 50 mM NaCl, 2 mM DTT, 0.5% (v/v) Triton X-100, and 6 mM MnCl₂. After 30 min of incubation at room temperature, 200 mM UTP, 1.5 mM CTP, 1.5 mM GTP, and 165 nM of $[\alpha\text{-}^{32}P]$-ATP (3000 Ci/mmol) were added into the mixture to initiate the reaction. The reaction mixtures (10 μl) were incubated at 30 °C for 3 h followed by 3 min at 70 °C and cooled down on ice, then stopped by the addition of 10 μl RNA Gel Loading Dye (2X) (Thermo Scientific) which contained the denaturing agent formamide. 10 μl reaction products were resolved on denaturing 7 M urea 20% polyacrylamide gel electrophoresis in TBE buffer, and analyzed by autoradiography through phosphorimaging (Typhoon FLA 7000, GE Healthcare Bio-Sciences). The sizes of the products were determined by comparing with a mixture of various lengths of marker RNA (poly ATP) labeled by T4 polynucleotide kinase (PNK) (New England Biolabs) using $[\gamma\text{-}^{32}P]$-ATP (3000 Ci/mmol). To investigate the effect of metal ions on enzymatic activity, MgCl₂, MnCl₂, ZnCl₂, and EDTA are each added to the reaction mixture at a final concentration of 5 mM. These assays were performed independently twice to confirm the results.

*Figure preparation*. All the representing model and electron density maps were generated using COOT[21], UCSF Chimera[46], and PyMOL[48]. Clustal X[49] and ESPript[50] were used to align multiple sequences.

**Reporting summary**. Further information on research design is available in the Nature Research Reporting Summary linked to this article.

## Data availability
The structure of JUNV L–Z complex has been deposited at the Protein Data Bank (PDB), (accession code: 7EJU). The cryo-EM density map of JUNV L–Z complex has been deposited at the Electron Microscopy Data Bank (accession code: EMD-31163). Source data are provided with this paper.

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

## Acknowledgements

The authors thank Boling Zhu, Lihong Chen and Xiaojun Huang for cryo-EM data collection, the Center for Biological imaging (CBI) in Institute of Biophysics (IBP) for EM work. We thank Hongjie Zhang at IBP for the guidance in handling radiolabeled chemicals. We thank Yuanyuan Chen, Zhenwei Yang, and Bingxue Zhou for assistance in MST interaction assay. We thank Bei Yang for assistance in cell biology experiments. We thank Xianjin Ou at IBP for the technical assistance in the fermentation/protein preparation. The research described in this article is supported by the National Basic Research Program (Grant No. 2017YFC0840300) to Y.C.; it is also supported by the National Key Research and Development Program (Grant No. 2020YFA0707503), the Strategic Priority Research Program of the Chinese Academy of Sciences (Grant No. XDB37030200), National Basic Research Program (Grant No. 2016DDJ1ZZ17) to X.L.

## Author contributions

H.K. and C.W. designed the study. J.C., H.K., and Y.X. performed protein purification and cryo-EM sample preparation. J.C., W.J., and Y.Q. performed the cryo-EM data collection. H.K., J.C., and C.W. conducted biochemical experiments and binding experiments; H.K., J.C., C.W., and Y.C. determined and refined the structure. H.K., J.C., and C.W. wrote the manuscript together with X.L. and Y.C. Z.R. supervised the project.

## Competing interests

The authors declare no competing interests.
