## [Peer Review File · Nature Communications]

REVIEWER COMMENTS

Reviewer #1 (Remarks to the Author):

This manuscript describes the structure of the Junin virus L-Z polymerase complex. The manuscript is very detailed in its description of the complex and interactions between L and Z but no images are shown that include the electron density to validate the accuracy of the model as it pertains to describing these interactions. The resolution (3.5 Å) is sufficiently high that many side chains should be visible for these interactions. This manuscript was quite difficult to read and needs significant editing.

Major comments:

The manuscript needs significant editing for grammar and readability.

Supp Fig 5E-F: Just because binding between vRNA and L is observed, doesn't mean the interaction is biologically relevant. We have seen binding between RNA and viral polymerases that doesn't appear to be a productive interaction. Is this interaction stable at higher salt concentrations and not just due to charge-charge interactions with a random surface on the polymerase? Buffer conditions for the EMSAs are not described in the methods.

Fig 5C: This seems much too speculative. What would induce the conformation change of the 680 loop to destabilize the interaction between L and Z? Why would it choose to undergo the reverse change during viral particle assembly? The description from lines 257-268 seems more plausible but is driven by the presence of Z, not intrinsic conformational changes in L as described for Fig 5C.

It's unclear how Z binding to the template exit channel inhibits L activity. Is this interaction strong enough to prevent displacement during RNA synthesis?

Minor comments:

Line 40: Abbreviations L and S refer to what? Long and short?

Line 53: Electron Microscopy should not be capitalized.

Line 57: "... host cell proteins, and..."

Line 75: "To improve the resolution, ..." It's unclear what the authors mean by "we merged 6,409 micrographs". Are these separate data sets or did they just use a subset for the initial 3D model and are now using the full data set? Total images collected is missing in the methods.

Line 78: "The final resolution of the map was 3.54 Å." This was mentioned in the last paragraph. Can omit here.

Line 78: The authors should comment on the similarity to MACV L protein to give the reader confidence in the use of it as an appropriate guide for model building.

Line 86: "comprising of flexible structures" needs to be rewritten for clarity.

Lines 90-107: This portion is very difficult to read as is.

Line 99-100: "The region encompassing the RING domain" → "The RING domain"

Line 105: Unclear what "conditions or factors favoring this are perhaps lacking" means.

Line 123: "forge" → "form"

Line 139: mainly

Line 141: Does Z not fold properly without L? Otherwise the statement that L partially helps to stabilize Z is not shown by the authors.

Line 142: "C-terminal of the RING domain" → "the C-terminus of the RING domain".

Line 168-170: This statement is not supported by the authors data. Dissociation constants don't inform at all that the interfaces the authors describe are those actually involved in the interaction. Mutagenesis of key residues would be more appropriate.

Line 190: Are zinc fingers known to interact with ssRNA?

Supp Fig 5F: JUNV is labeled twice. Is this correct? What is the difference?

In general supplemental figure captions are lacking in information.

Line 333: How is 362,000 particles 54.1% of the total 1,255,000 particles?

Line 346: "with rotamer" unclear what this means. How specifically was the Ramachandran plot used to guide validating the geometries of the model? Allowable Ramachandran values do not indicate that the model is accurately built.

Line 353: References 46 and 47 are not listed.

Reviewer #2 (Remarks to the Author):

Kang et al. described a 3.5 Å cryo-EM structure of Junin virus (JUNV) L protein in complex with Z. JUNV is a segmented negative-sense RNA virus, and many other L proteins from segmented and non-segmented negative-sense RNA viruses have been described previously. Kang et al. also compared and revealed a highly conserved architecture of the JUNV L protein with other L proteins. In this manuscript, Kang et al. demonstrated the function of L protein is regulated by the binding of Z protein at the RNA product exit site.

Overall, it is a paper with solid experimental data. The reported data unambiguously showed the conclusion. Kang et al. also proposed a model of Z as a switch to turn on/off the viral RNA synthesis via its interaction with L protein and highlighted the possible mechanism of JUNV transcription, replication, and regulation.

However, there are still some issues regarding the conclusion which need to be addressed:

1. Line 72, "Analysis of the purified L-Z complex using SDS-PAGE indicted that the complex contained the full-length L protein and Z protein". To confirm the identifies of the complex, it is best to do mass spec, but not use SDS-PAGE as proof.

2. Line 83, "a few gaps within the connecting loops". Please specify the residue number ranges, like for Z protein (lines 85-86).

3. Lines 104-106. The explanation for "A possible second zinc finger" is not clear. Did the authors observe the zinc finger?

4. Lines 115-116. "residue D1187 of motif A can bind" and "Motif D and motif E can form". Does "can" here mean the actual function of the residue or motifs or it just means speculation? Any proof?

5. Lines 165-168. It is a little odd that the binding affinity of full-length L to full-length Z (95.3 ± 52.8 nM) is weaker than the fragment of Z (AA48-78) 28.9 ± 17.6 nM. Please explain.

6. Lines 226-236. The reasoning in the discussion of the dimerization of the JUNV L needs some clarification. Kang et al. described the preparation of the JUNV L without observing the dimeric state. The question is: does the author believe dimerization is required for the JUNV L function in viral replication?

7. Lines 254-268. "the plausible model for the regulation of JUNV L protein during the infection cycle". It is good that such a model can be proposed for several steps. Please list relevant references for each step. If no strong support for a certain step, please consider removing it or only propose what the structures and biochemical data can support.

8. Figure 1. Panel b. What is the molar ratio of L:Z? It is understood that Z is a much smaller protein.

9. Figure 2. The color of Z (dark blue) is very similar to the Endonuclease domain of L (blue). Any way to improve the contrast?

10. Figure 3. The color of Z protein keeps changing and confusing, dark blue in panel A and red in Panel B. Can the author make it consistent since Z is a key feature here?

11. Figure 4. What is the relationship between panels b and c?

12. Figure 5. 680-loop comparison is good. Can the author highlight the conserved residues of the JUNV L in Panel C?

13. Figure S5. Panel e. The interpretation of the results is not complete. Not sure what to expect?

Reviewer #1 (Remarks to the Author):

This manuscript describes the structure of the Junin virus L-Z polymerase complex. The manuscript is very detailed in its description of the complex and interactions between L and Z but no images are shown that include the electron density to validate the accuracy of the model as it pertains to describing these interactions. The resolution (3.5 Å) is sufficiently high that many side chains should be visible for these interactions. This manuscript was quite difficult to read and needs significant editing.

Our response: We thank the reviewer for the constructive suggestion to show electron density of the interaction between JUNV L and Z. We have included the electron density map for side chains at L-Z binding interface in supp figure 5a-b to illustrate the validity of local resolution.

Major comments:

The manuscript needs significant editing for grammar and readability.

Our response: We thank the reviewer for this suggestion. The manuscript has now been revised by a native English speaker with experience in structural biology.

Supp Fig 5E-F: Just because binding between vRNA and L is observed, doesn't mean the interaction is biologically relevant. We have seen binding between RNA and viral polymerases that doesn't appear to be a productive interaction. Is this interaction stable at higher salt concentrations and not just due to charge-charge interactions with a random surface on the polymerase? Buffer conditions for the EMSAs are not described in the methods.

Our response: We partially agree with the reviewer in this point. Analysis of JUNV L RdRp active site showed that the interaction between L and RNA strands should be driven by electrostatic interaction, which is probably compromised at high salt concentrations. However, the buffer we used in the EMSA assay contains NaCl at 500 mM, a relatively high concentration, the interaction was still affirmed, indicating that JUNV L and 3'-vRNA interact specifically. Also, as suggested by the reviewer, details of buffer conditions have been included in the method section.

Fig 5C: This seems much too speculative. What would induce the conformation change of the 680 loop to destabilize the interaction between L and Z? Why would it choose to undergo the reverse change during viral particle assembly? The description from lines 257-268 seems more plausible but is driven by the presence of Z, not intrinsic conformational changes in L as described for Fig 5C.

Our response: We thank the reviewer for this constructive suggestion. We agree that 680 loop does not initiate the conformational change to mediate or regulate L-Z association/dissociation. Currently there is little published data as to what causes the conformational change on 680 loop to repel the Z protein. However, previous results clearly showed that Z protein is able to bind a series of host factors e.g. eIF4E (*Campbell et.al, J Virology 2000,74, 3293-3300*) via the RING domain which is responsible for interaction of Z with L. We hypothesize that host factors induce Z protein conformation or alternatively competitively bind Z, resulting in dissociation of Z from L.

As to why L would undergo the reverse change during viral particle assembly, we think this change is passive, caused by the binding of Z during virion assembly when Z begins to accumulate in host cell. Z plays the major role in conformational change induction of L 680 loop on structural comparison with MACV apo L protein, as well as low-resolution cryo-EM density of JUNV apo L (~4.0Å resolution, unpublished data).

We thank the reviewer for positive comment on explanation with lines 257-268. We have rephrased the description for explaining the role of loop 680.

It's unclear how Z binding to the template exit channel inhibits L activity. Is this interaction strong enough to prevent displacement during RNA synthesis?

Our response: Our writing might have cause a slight misunderstanding related to the binding location of Z on L. According to our solved JUNV L structure and comparison with MACV L, the Z protein binding site is at the nascent RNA exit, which blocks the nascent 5' RNA from leaving and therefore inhibits both transcription and replication of the RNA genome. The MST binding assay showed the K_D of native L and Z is ~393nM, which is relatively strong to prevent the exit of product RNA.

Minor comments:

Line 40: Abbreviations L and S refer to what? Long and short?

Our response: L and S refer to “long” and “short”, respectively. We have rephrased the sentence and included the abbreviations for clarity.

Line 53: Electron Microscopy should not be capitalized.

Our response: We have changed the “Electron Microscopy” to “electron microscope”.

Line 57: “... host cell proteins, and...”

Our response: We have added the “,” as suggested by the reviewer.

Line 75: “To improve the resolution, ...” It's unclear what the authors mean by “we merged 6,409 micrographs”. Are these separate data sets or did they just use a subset for the initial 3D model and are now using the full data set? Total images collected is missing in the methods.

Our response: The 6409 micrographs were the total number from 3 separate data sets. We have rephrased the sentence to “To improve the resolution, we merged 6,409 micrographs from 3 separate data sets, and obtained a final 3.54 Å map...”. We also added sentence “We collected a total of 6,409 images, of which 2,184 were selected for the calculation of the initial model.” at Method “Image processing” section.

Line 78: “The final resolution of the map was 3.54Å.” This was mentioned in the last paragraph. Can omit here.

Our response: We agreed with the reviewer's comment and have deleted this sentence.

Line 78: The authors should comment on the similarity to MACV L protein to give the reader

confidence in the use of it as an appropriate guide for model building.

Our response: We thank the reviewer for this suggestion. We have rephrased these sentences to include the amino acid sequence identity of 73%, and r.m.s.d. of C α atoms at 0.95Å with MACV L, in order to show the overall structural similarity.

Line 86: “comprising of flexible structures” needs to be rewritten for clarity.

Our response: We have rewritten the sentence to “...,while N-terminal (AA 1-30) and C-terminal (AA 83-94) regions was not modeled due to the flexible nature and missing density.” for clarity.

Lines 90-107: This portion is very difficult to read as is.

Our response: We have rephrased this paragraph, and improved its readability and clarity.

Line 99-100: “The region encompassing the RING domain” —> “The RING domain”

Our response: We agree with the reviewer and deleted “The region encompassing” from the sentence to make it more concise.

Line 105: Unclear what “conditions or factors favoring this are perhaps lacking” means.

Our response: We have modified the sentences to “However, distribution and orientation of the residues were distorted in the structure, and we did not identify density for zinc ion.”

Line 123: “forge” —> “form”

Our response: We have replaced “forge” with “form”, as suggested by the reviewer.

Line 139: mainly

Our response: We have rephrased the sentence to “The core lobe domain of JUNV L protein interacts with the RING domain of Z protein.”.

Line 141: Does Z not fold properly without L? Otherwise the statement that L partially helps to stabilize Z is not shown by the authors.

Our response: According to our experiment result, Z protein could fold properly without L. Therefore, we have deleted the description of “partially to help the latter to stabilize its structure”.

Line 142: “C-terminal of the RING domain” —> “the C-terminus of the RING domain”.

Our response: We have changed the phrase as suggested by the reviewer.

Line 168-170: This statement is not supported by the authors data. Dissociation constants don't inform at all that the interfaces the authors describe are those actually involved in the interaction. Mutagenesis of key residues would be more appropriate.

Our response: We agree with the reviewer on this point. The Z-derived peptides (aa18-47 and aa48-78, as indicated in figure 4) were chosen based on the complex interaction interface and showed similar K_D as native Z, suggesting they were the main binding region. We also have performed mutagenesis of residues that participate in L-Z interaction (L mutants: F1377A,

VNN(1177-1179)AAA, F688A, Y687A&F1377A, F688A&F1377A; Z mutants: W43A, R36A, W43A&R36A) and tested the affinity of mutants with native binding partner(included in sup figure 6). However, none of the mutants could totally abolish the binding ability, except L-Y687A&F1377A and L-F688A&F1377A. Therefore, we deduce that the interaction occurs at multiple sites, and single point mutations are not sufficient to abolish the L-Z interaction. Therefore, we have added mutants binding result and rephrased the paragraph.

Line 190: Are zinc fingers known to interact with ssRNA?

Our response: There are reported examples (J Mol Biol. 2011 Mar 25;407(2):273-83. Nucleic Acids Res. 2016 Nov 2;44(19):9153-9165.) of zinc finger-containing proteins that are known to interact with ssRNA. We have updated these references to our manuscript as well.

Supp Fig 5F: JUNV is labeled twice. Is this correct? What is the difference?

In general supplemental figure captions are lacking in information.

Our response: It is correct that the two JUNV L labels are indicators for monomeric and dimeric forms of L protein on NuPage. For simplicity, we have reduced the number of labels to one in supp Fig 4c.

Also, we have rephrased and modified the figure captions, to make them more informative.

Line 333: How is 362,000 particles 54.1% of the total 1,255,000 particles?

Our response: We thank the reviewer for pointing the error, we have corrected the number to 28.8% in the sentence.

Line 346: “with rotamer” unclear what this means. How specifically was the Ramachandran plot used to guide validating the geometries of the model? Allowable Ramachandran values do not indicate that the model is accurately built.

Our response: We thank the reviewer for pointing out the vagueness here. We have now rewritten the sentence to “We used PHENIX to calculate the structure factors. The complete structure was subjected to global real-space refinement in multiple cycles using rotamer and Ramachandran plot restraints.”.

Line 353: References 46 and 47 are not listed.

Our response: We thank the reviewer to pointing the missing references. We have now added these two references to the revised manuscript at the original site.

Reviewer #2 (Remarks to the Author):

Kang et al. described a 3.5Å cryo-EM structure of Junin virus (JUNV) L protein in complex with Z. JUNV is a segmented negative-sense RNA virus, and many other L proteins from segmented and non-segmented negative-sense RNA viruses have been described previously. Kang et al. also compared and revealed a highly conserved architecture of the JUNV L protein

with other L proteins. In this manuscript, Kang et al. demonstrated the function of L protein is regulated by the binding of Z protein at the RNA product exit site.

Our response: We thank the reviewer for summarizing so clearly our main findings (no further response is needed here).

Overall, it is a paper with solid experimental data. The reported data unambiguously showed the conclusion. Kang et al. also proposed a model of Z as a switch to turn on/off the viral RNA synthesis via its interaction with L protein and highlighted the possible mechanism of JUNV transcription, replication, and regulation.

Our response: We thank the reviewer for these positive comments.

However, there are still some issues regarding the conclusion which need to be addressed:

1. Line 72, “Analysis of the purified L-Z complex using SDS-PAGE indicted that the complex contained the full-length L protein and Z protein”. To confirm the identifies of the complex, it is best to do mass spec, but not use SDS-PAGE as proof.

Our response: Apart from SDS-PAGE, we also have sent the bands on SDS-PAGE for peptide mass fingerprinting using MALDI-TOF MS after tryptic in-gel digestion. Both L and Z proteins have been identified and confirmed.

2. Line 83, “a few gaps within the connecting loops”. Please specify the residue number ranges, like for Z protein (lines 85-86).

Our response: We thank the reviewer for the reminding. The gaps within loops that were missing from the structure were summarized in supp Table. 2. We have added the reference to supp Table 2.

3. Lines 104-106. The explanation for “A possible second zinc finger” is not clear. Did the authors observe the zinc finger?

Our response: It was shown that Z proteins from other negative-strand RNA virus, e.g. Lassa virus, contains two zinc fingers. Apart from an affirmed zinc finger(C41-C42-C58-C61) with density for zinc ion, we also found 4 amino acids(C52-H55-C72-C75) in vicinity of similar location but the distribution and orientation of the residues were distorted, and no density for zinc ion was discerned. Therefore, we referred to it as “another plausible zinc finger” since it might form a standard zinc finger structure in certain situation. We also have rephrased the sentences in the paragraph to make it clear.

4. Lines 115-116. “residue D1187 of motif A can bind” and “Motif D and motif E can form”. Does “can” here mean the actual function of the residue or motifs or it just means speculation? Any proof?

Our response: We agree with the review that the word “can” here brought some vagueness to the understanding. To answer the reviewer’s questions, D1187 is actually bound with divalent cation, a Mg^{2+} ion, with clear density map, which is also highly conserved in other negative viruses. On the other hand, Motif D and motif E are in close contact, which is bridged by interaction with the Mg^{2+} ion, as shown in the structure. We have rephrased the sentences to

more clearly describe the motif features.

5. Lines 165-168. It is a little odd that the binding affinity of full-length L to full-length Z (95.3±52.8 nM) is weaker than the fragment of Z (AA48-78) 28.9±17.6nM. Please explain.

Our response: We thank the reviewer for pointing this out. We have recently re-performed the MST binding assay, with freshly prepared L-*wt* at a higher purity and data repetition of at least thrice. The new MST binding assay results (supp figure 6 a-c) indicated that the full-length Z binds L protein with an higher affinity of 2-3 fold compared with two peptides, which is reasonable.

6. Lines 226-236. The reasoning in the discussion of the dimerization of the JUNV L needs some clarification. Kang et al. described the preparation of the JUNV L without observing the dimeric state. The question is: does the author believe dimerization is required for the JUNV L function in viral replication?

Our response: We thank the reviewer for the constructive suggestion; we have now made modifications to this part. To answer the reviewer's question, we do not think dimerization is a prerequisite for RNA replication, since we detected only L monomer in solution by gel filtration as well as in cryo-EM particle 2D classification, and the monomeric L protein demonstrated *in vitro* RNA replication activity. However, we deduce that dimerization of JUNV L might play the role in replication/transcription conversion, as has been suggested in other negative RNA virus such as LCMV and MACV.

7. Lines 254-268. "the plausible model for the regulation of JUNV L protein during the infection cycle". It is good that such a model can be proposed for several steps. Please list relevant references for each step. If no strong support for a certain step, please consider removing it or only propose what the structures and biochemical data can support.

Our response: We thank the reviewer for the positive comment on the model proposition, and we totally agree with the reviewer that references should be added as support of individual steps. We have now added references to the individual steps to provide a solid scientific background, and we also have rephrased the description of the model. We found that our original model might have over-stressed the role of 680 loop in the modulation of JUNV L function. The "molecular switch" effect might have been triggered by host factors that have been confirmed to interact with the Z protein, while their roles in the viral RNA transcription/replication have not been elucidated yet.

8. Figure 1. Panel b. What is the molar ratio of L:Z? It is understood that Z is a much smaller protein.

Our response: The molar ratio of L:Z in gel filtration and SDS-PAGE of Panel b is 1:1, as is shown clearly in the cryo-EM map (the sample shown in panel b is the same sample applied to cryoEM sample preparation), with one L protein and one Z protein.

9. Figure 2. The color of Z (dark blue) is very similar to the Endonuclease domain of L (blue). Any way to improve the contrast?

Our response: We agree with the reviewer for the resemblance of color between Z and

Endonuclease domain. We have now modified the color of Endonuclease domain to light-purple and kept the Z protein in former color to make a better illustration and clearer contrast.

10. Figure 3. The color of Z protein keeps changing and confusing, dark blue in panel A and red in Panel B. Can the author make it consistent since Z is a key feature here?

Our response: We thank the reviewer for the constructive suggestion. We have now modified the color in figures containing Z protein in our revised version, to ensure that all Z proteins are presented in consistent color of dark or bright blue.

11. Figure 4. What is the relationship between panels b and c?

Our response: The two major regions of interaction site between L and Z, which is shown as a whole view in panel a, are separately shown as close-in view in panels b (left region) and c (right region) for a better illustration.

12. Figure 5. 680-loop comparison is good. Can the author highlight the conserved residues of the JUNV L in Panel C?

Our response: We thank the reviewer for the positive comment. The conserved residues of JUNV L (I677, L678 and A684) could not be shown clearly in panel C since they would be overlapping with the existing structure. Therefore, we choose to highlight these residues in panel D instead.

13. Figure S5. Panel e. The interpretation of the results is not complete. Not sure what to expect?

Our response: We thank the reviewer for pointing this out. We have now added the description of polymerase result in the figure legend of our revised version: “Mg²⁺ and Mn²⁺ were found to promote RNA replication activity, while Zn²⁺ can not activate L protein.”

REVIEWER COMMENTS

Reviewer #1 (Remarks to the Author):

The authors have addressed all my concerns.

Reviewer #2 (Remarks to the Author):

I want to thank the authors for addressing my initial comments. Following the revision to the article, I do not have more questions now.

Reviewer #1 (Remarks to the Author):

The authors have addressed all my concerns.

Our response: We thank the reviewer for the positive response.

Reviewer #2 (Remarks to the Author):

I want to thank the authors for addressing my initial comments. Following the revision to the article, I do not have more questions now.

Our response: We thank the reviewer for the positive response.